# Investigation of the Stability and Hydrogen Evolution Activity of Dual-Atom Catalysts on Nitrogen-Doped Graphene

**DOI:** 10.3390/nano12152557

**Published:** 2022-07-25

**Authors:** Qiansong Zhou, Meng Zhang, Beien Zhu, Yi Gao

**Affiliations:** 1Key Laboratory of Interfacial Physics and Technology, Shanghai Institute of Applied Physics, Chinese Academy of Sciences, Shanghai 201800, China; zhouqiansong@sinap.ac.cn; 2University of Chinese Academy of Sciences, Beijing 100049, China; 3Department of Physics, School of Physical Science and Technology, Ningbo University, Ningbo 315211, China; zhangmeng733@126.com; 4Interdisciplinary Research Center, Zhangjiang Laboratory, Shanghai Advanced Research Institute, Chinese Academy of Sciences, Shanghai 201210, China

**Keywords:** graphene, transition metal atoms, hydrogen evolution reaction, DFT

## Abstract

Single atom catalysts (SACs) have received a lot of attention in recent years for their high catalytic activity, selectivity, and atomic utilization rates. Two-dimensional N-doped graphene has been widely used to stabilize transition metal (TM) SACs in many reactions. However, the anchored SAC could lose its activity because of the too strong metal-N interaction. Alternatively, we studied the stability and activity of dual-atom catalysts (DACs) for 24 TMs on N-doped graphene, which kept the dispersion state but had different electronic structures from SACs. Our results show that seven DACs can be formed directly compared to the SACs. The others can form stably when the number of TMs is slightly larger than the number of vacancies. We further show that some of the DACs present better catalytic activities in hydrogen evolution reaction (HER) than the corresponding SACs, which can be attributed to the optimal charge transfer that is tuned by the additional atom. After the screening, the DAC of Re is identified as the most promising catalyst for HER. This study provides useful information for designing atomically-dispersed catalysts on N−doped graphene beyond SACs.

## 1. Introduction

Single atom catalysts (SACs) exhibit the maximum atomic efficiency, high selectivity, and high activity towards a variety of chemical reactions, which opens up a new frontier in the field of catalysis [1,2,3,4]. However, the practical chemical processes require a huge number of SACs, which make them easy to aggregate and form clusters owing to their high surface energies [5]. It is well known that the rational design of catalysts is a key step in catalysis research [6]. As a result, some substrates are introduced to attach single metal atoms and prevent the formation of metal clusters [7], such as Pt_1_/FeO_X_ [8], Fe_1_/SiO_2_ [9], and Pt_1_/C, [10]. Among them, two-dimensional defected graphene is used as the effective template to stabilize the SACs by taking advantage of a large surface area, good conductivity, good stability, and strong dispersibility [11,12,13,14]. Recently, a synergy strategy of co-doping graphene with transition-metal (TM) and nitrogen (N) atoms has been proposed to enhance the surface stability and catalytic activity simultaneously. For example, Wang et al. reported Fe SACs with triple nitrogen coordination (Fe−N_3_) on graphene has excellent nitrogen reduction reaction activity [15]. Sa et al. synthesized single atom Ni anchored in N−doped graphene, which gave high performance for the carbon dioxide reduction reaction due to the dispersed highly catalytically active sites [16]. Zhong et al. studied the oxygen reduction reaction mechanism of TMs (Cr, Mn, Fe, and Co) with quadruple-coordinated nitrogen (TM−N_4_) on graphene using density functional theory (DFT) calculations and discovered that Fe/N/C had the best ORR activity [17]. Cheng et al. discovered that the single platinum atom catalysts exhibit significantly enhanced catalytic activity of the hydrogen evolution reaction, and Fang et al. uncovered that the near-free single-atom Pt possesses the favorable bonding energies with the reactants that are responsible for the superior hydrogen evolution reaction (HER) activity [18,19]. Hossain et al. and Jung et al. confirmed the excellent HER activity of single-atom Co on nitrogen-doped graphene from the perspectives of theoretical design and experimental verification [20,21]. Zang et al. demonstrated that atomically-dispersed Ni with triple nitrogen coordination (Ni−N_3_) can achieve efficient hydrogen evolution reaction (HER) performance in alkaline media, which was attributed to the lower coordination number than Ni−N_4_ to facilitate water dissociation and hydrogen adsorption [22]. 

Despite the significant progress in theory and experiments, the stability of anchored SACs possibly comes with the sacrifice of activity because of the too strong metal-support interaction. Compared with SACs, dual-atom catalysts (DACs) have emerged as a novel frontier because the synergistic effect between adjacent metal atoms can promote their catalytic activity [23]. DACs on the N−doped graphene can also be highly dispersed but have different electronic structures from SACs [24]. As the single atoms in SACs always have a stronger adsorption capacity than any other sites of support, the secondary atom is thermodynamically more likely to be captured to form DACs [25]. Therefore, DACs could form simultaneously with SACs if the practical loading of metal atoms on the surface is larger than the number of vacancies (overloading). Several studies have confirmed that using nitrogen-doped graphene as a substrate, DACs can serve as good electrocatalysts in HER [26,27,28]. However, there are limited studies of DACs on the N-doped graphene in the literature with respect to the studies of SACs. 

In this work, we performed DFT calculations to study the stability of the TM atoms on the two-dimensional 3N−doped single-vacancy graphene (*SVGN*_3_). A total of 24 TM elements were systematically investigated, including the third period (Ti, V, Cr, Mn, Fe, Co, Ni, and Cu), fourth period (Zr, Nb, Mo, Tc, Ru, Rh, Pd, and Ag), and fifth period (Hf, Ta, W, Re, Os, Ir, Pt, and Au). Among the 24 TM elements, 17 of them generally prefer to form the anchored SACs (*TM*_1_−*SVGN*_3_), while the rest of them are prone to forming DACs (*TM*_2_−*SVGN*_3_) instead of SACs. When the TM atoms are overloaded, all *TM*_2_−*SVGN*_3_ become stable. Then, the catalytic activities of SACs/DACs in the HER were predicted based on the Gibbs free energy of H adsorption [29,30]. We found 13 DACs showed higher HER activity than SACs, in which Re_2_−*SVGN*_3_ show the superior activity.

## 2. Materials

The spin-polarized Density Functional Theory (DFT) calculations were carried out using the exchange-correlation potential that was prescribed by Perdew–Burke–Ernzerhof (PBE) [31] and projector augmented wave (PAW) [32] methods within the Vienna ab initio simulation package (VASP) [33]. We also tested RPBE and PBE+D3 functionals with 3d elements and found that these functionals gave similar results (details in Appendix A). The plane-wave basis set was given a kinetic energy cutoff of 600 eV. The total energies were converged to 10^−6^ eV, and forces criteria was 0.02 eV/Å. The geometric optimization was performed using a Monkhorst–Pack grid of 3 × 3 × 1 k-points [34]. The Climbing image Nudged Elastic Band (CI−NEB) method was used to study the minimum energy path for dual-atom migration [35]. The calculated lattice parameter of graphene is 2.47 Å, which agreed with the reported theoretical and experimental results [36,37]. The optimized bulk graphene C–C bonds are 1.43 Å and the graphene supercell is then constructed based on the calculated lattice constant. In the xy plane, a 6 × 6 supercell (72 atoms) was used to eliminate the effects from the periodic structure. The distance between the graphene monolayer and its mirror copies was adjusted at 15Å in the z-axis, which was enough to preclude interactions between the two monolayers. In this work, we focus on *SVGN*_3_ as the substrate because of its applications in experiments and the lower formation energy of single-vacancy than that of the double-vacancy, which indicates the possibility of a single-vacancy site is high while synthesizing defect-containing graphene substrates (details in Appendix A) [22,38].

The binding energy ($E_{b}^{n}$ or $E_{b}^{G}$) of the *TM*_n_ on *SVGN*_3_ substrate or pure graphene (G) substrate are calculated as below:(1)$$\begin{array}{c} E_{b}^{n}=E_{TM_{n}-SVGN_{3}}-E_{SVGN_{3}}-nE_{TM_{1}}\;\left(n=1\;or\;2\right) \end{array}$$
(2)$$\begin{array}{c} E_{b}^{G}=E_{TM_{1}-G}-E_{G}-E_{TM_{1}} \end{array}$$
where $E_{TM_{n}-SVGN_{3}}$ and $E_{TM_{1}-G}$ are the total energies of *TM*_n_−*SVGN*_3_ and *TM*_1_−*G* slab, respectively. $E_{SVGN_{3}}$ is the energy of *SVGN*_3_, $E_{G}$ is the energy of G, $E_{TM_{1}}$ is the energy of an isolated transition metal atom in the vacuum, and *TM*_1_ and *TM*_2_ stand for single-atom and dual-atom, respectively.

The adsorption energy $∆E_{H^{*}}$ of H atom was calculated by: (3)∆EH*=E(TMn−SVGN3+H)−ETMn−SVGN3−12EH2 (n=1 or 2)
where $E_{\left(TM_{n}-SVGN_{3}+H\right)}$ represents the total energy of the *TM*_n_−*SVGN*_3_ systems with one adsorbed H atom, and $E_{H_{2}}$ represents the energy of a gas phase $H_{2}$ molecule.

Under standard conditions, the Gibbs free energy of H adsorption, $∆G_{H^{*}}$ was obtained by the equation:(4)$$\begin{array}{c} ∆G_{H^{*}}=∆E_{H^{*}}+∆E_{ZPE}-T∆S_{H^{*}} \end{array}$$
where $∆E_{ZPE}$ corresponds to the zero-point energy of adsorbed hydrogen and hydrogen in the gas phase $H_{2}$ molecule, and $∆S_{H^{*}}$ is the entropy difference between the adsorbed state and gas phase. The temperature (T) is 298.15 K. In this work, the $E_{ZPE}$ and $S_{H^{*}}$ values were calculated based on the vibrational frequencies through the VASPKIT [39].

## 3. Results and Discussion

### 3.1. Geometric Structures 

#### 3.1.1. Single-Atom

In Figure 1, considering the symmetry of the structure, we marked the possible adsorption sites on G and *SVGN*_3_, respectively. As consistent with previous reports, the calculated C-C bond length on the optimized G is 1.42 Å, while the calculated bond lengths of C-C and C-N in the pore on the optimized *SVGN*_3_ are 1.34 Å and 1.45 Å, respectively [40,41]. There are three adsorption sites on G: hollow (H), bridge (B), and top (T), and five adsorption sites on *SVGN*_3_: single vacancy (S), top position of nitrogen (a), bridge position between nitrogen and nitrogen, (B’) and different top positions of carbon (b or c) on the substrate.

Table 1 and Table 2 give the geometric parameters and binding energies of the most stable adsorption site for *TM*_1_−*G* and *TM*_1_−*SVGN*_3_, respectively, where M1 is a TM atom in these systems. When the M1 is adsorbed on G, the H site is the most stable adsorption site for 13 elements (Ti, V, Fe, Co, Ni, Zr, Nb, Tc, Ru, Rh, Hf, Ta, and Os); B site is the most stable adsorption site for 7 elements (Cr, Mo, Pd, Ag, W, Ir, and Pt); and the remaining elements (Mn, Cu, Re, and Au) sit on T site. Generally speaking, when the M1 is chemically adsorbed, it tends to be adsorbed at the H site, while in the case of physical adsorption, the coordination number of the M1 atom with C atoms on the surface of pure graphene decreases (H→B→T). For *TM*_1_−*SVGN*_3_, the M1 is embedded in the position of the S point for all the elements. Comparing Table 1 and Table 2, for elements (Cr, Mo, Pd, Ag, W, Ir, and Pt) whose B site is the best adsorption site on pure graphene, the value difference between $E_{b}^{1}$ and $E_{b}^{G}$ of Pd(Pt) is relatively small, making the C-C bonds (B site) and the single-defect vacancy (S site) to single atom in Pd−*SVGN*_3_(Pt−*SVGN*_3_) competitive for the adsorption of a single atom, eventually resulting in one d_M1−Nx_ that is much larger than the other two. All the results are consistent with previous works [40,42,43,44]. As shown in Figure 2, we found much stronger $E_{b}^{1}$ than $E_{b}^{G}$, which is in line with the expectation. In addition, the binding energy of the single atom on the *SVGN*_3_ substrate is smaller than that on the single-vacancy substrate without an N atom [14]. The binding energy is roughly positively associated with the height (h) from the M1 to the pure graphene in the *TM*_1_−*G* and with the average distance (d_M1−Nx_) from the anchored M1 to the N atom in the *TM*_1_−*SVGN*_3_ (details in Appendix A).

#### 3.1.2. Dual-Atom

For dual-atoms, we defined M1 as the TM anchored at the vacancy and M2 as the additionally attached one and six initial configurations that were sampled: (Figure 3Ⅰ) M2 is also at the vacancy site but on the different side of the graphene sheet; (Figure 3Ⅱ) M2 is at position a, (Figure 3Ⅲ) M2 is at the position b, (Figure 3Ⅳ) M2 is at the position c, (Figure 3Ⅴ) M1 is at B’, while M2 is at a (Figure 3Ⅵ), both atoms stay on the same side of the graphene sheet and perpendicular to the graphene sheet.

The structural and energetic parameters of DACs are summarized in Table 3. We found that there is no optimum configuration matching to Figure 3Ⅲ for all the DACs. First, according to Figure 2, in the same period, the $E_{b}^{1}$ value of the IVB elements is the largest, so that the overloaded metal atom may also tend to bond with the single-defect vacancy, and finally the *TM*_2_ of the group IVB elements (Ti, Zr, and Hf) takes the stable configuration as Ⅰ. Second, the V_2_, Nb_2_, Rh_2_, Pd_2_, Ta_2_, and W_2_ are all on the same side of the *SVGN*_3_. Among them, V_2_−*SVGN*_3_, Nb_2_−*SVGN*_3_, and Ta_2_−*SVGN*_3_ prefer to adopt the Ⅱ configuration, whereas the configurations of Rh_2_−*SVGN*_3_ and Pd_2_−*SVGN*_3_ are Ⅳ, and W_2_−*SVGN*_3_ forms a stable structure as Ⅴ. Finally, the *TM*_2_ of remaining 15 elements (Cr, Mn, Fe, Co, Ni, Cu, Mo, Tc, Ru, Ag, Re, Os, Ir, Pt, and Au) are perpendicularly adsorbed on *SVGN*_3_ (configuration Ⅵ). Additionally, in the *TM*_2_−*SVGN*_3_ system (details in Appendix A), the average distance (d_M1−Nx_ and d_M2−Nx_) between the dual-atoms (M1 and M2) and the N atom is related positively to the binding energy. We also tested other possible configurations and discovered that they are all unstable.

### 3.2. Stability

To evaluate the relative stability between SACs and DACs, the difference of the binding energy was calculated using the following equation:(5)$$\begin{array}{c} ∆E_{1}=E_{b}^{2}-2*E_{b}^{1} \end{array}$$

When $∆E_{1}$ is negative, the anchored DAC is energetically more stable than two isolated SACs that are trapped by *SVGN*_3_. As shown in Figure 4a, all the isolated SACs of 3d and 4d elements are more stable except for Mo and Tc ($∆E_{1}$ is −0.84 eV and −2.72 eV, respectively). Among the 5d elements, the late TM elements (Re, Os, Ir, Pt, and Au) tend to form DACs instead of being dispersed into SACs, with $∆E_{1}$ of −0.50, −0.61, −1.25, −1.31, and −1.34 eV, respectively. It is interesting to notice that when the number of 5d valence electrons (from 5d^5^ to 5d^10^) increases, the $∆E_{1}$ values of these elements (Re, Os, Ir, Pt, and Au) gradually become more negative, which strongly show the DACs of these elements are more prone to forming than SACs. The investigation of the catalytic properties of these DACs are necessary.

For other elements, $∆E_{1}$ is positive, which means the isolated SACs are preferred to form on the *SVGN*_3_ when there are enough vacancies. However, as we mentioned in the Introduction, the number of TM atoms could be slightly larger than the number of vacancies in practical synthesis. When the TM atoms are overloaded, we consider the extra atoms can either disperse on the graphene surface or attach to the trapped SACs to form DACs. The relative stability was evaluated as: (6)$$\begin{array}{c} ∆E_{2}=E_{b}^{2}-E_{b}^{1}-E_{b}^{G} \end{array}$$

If $∆E_{2}$ is negative, the DACs structure is more stable. The calculated values of $∆E_{2}$ range from −0.55 to −6.63 eV (Figure 4b), showing the relative stability of *TM*_2_−*SVGN*_3_. The absolute value of $∆E_{2}$ generally shows that it first decreases and then increases with the increase of the atomic number of TM for the 3d and 4d elements. The more negative the value of $∆E_{2}$, the better the stability of *TM*_2_−*SVGN*_3_. As is known, the surface energy of TM atoms will first increase and then decrease as the atomic number increases [45]. The elements in the middle of the period have more positive surface energies than the early and late TM elements in the same period [46,47], meaning the metal–metal interaction is stronger, and M2 is more likely to bond with M1 to form DACs. For example, in the 5th period elements, W, Re, and Os have the larger surface energies than the other elements, while in our calculations, W_2_−, Re_2_−, and Os_2_−*SVGN*_3_ possess higher relative stabilities. DACs stability can also be considered in terms of TM atoms migration energy barriers, for example (Figure 5), Re_2_ migration energy barrier in Re_2_−*SVGN*_3_ is 3.16eV, further confirming the excellent stability of Re_2_−*SVGN*_3_.

### 3.3. HER Activity

With the growing worldwide energy demand and increasing environmental pressures, producing high-purity hydrogen from abundant water via an electrochemical HER as a clean energy source is a sustainable and cost-effective option [48]. $∆G_{H^{*}}$ is well-documented as a singular activity descriptor of the HER, and there is a reported relationship between $∆G_{H^{*}}$ and HER activity in the system of graphene-supported single atom catalysts [20]. According to the Sabatier principle, the closer $∆G_{H^{*}}$ to zero (|$∆G_{H^{*}}$| ≤ 0.10), the higher catalytic activity of HER [29,30,49]. If the interaction between H and the catalyst is too weak ($∆G_{H^{*}}$ ≫ 0), hydrogen adsorption (Volmer reaction: $H^{+}+e^{-}+*\to H^{*}$, where * refers to catalysts surface) will be limited, while too strong interaction ($∆G_{H^{*}}$≪ 0) creates difficulty for the desorption step (Tafel or Heyrovsky step: $H^{*}+H^{*}\to H_{2}+2^{*}$ or $H^{+}+H^{*}+e^{-}\to H_{2}+*$) to proceed on the catalyst surface [50,51]. Based on these results, we calculated $∆G_{H^{*}}$ by considering the H adsorption on the stable SACs (*TM*_1_−*SVGN*_3_)/DACs (*TM*_2_−*SVGN*_3_) models.

The active site for HER is highly related to the adsorption site of H atom. We tested several possible configurations for the H atom adsorbed on SACs and DACs, and obtained the optimal configurations: For SACs, the most favorable adsorption sites of H is on top of the TM atoms; for DACs, the best adsorption sites of H are divided into two different situations: (1) the H bonds to the second TM atom in Ti, Cr, Zr, Mo, Tc, Ru, Os, Ir, Pt, and Hf DACs; (2) it adsorbs between the two TM atoms in V, Mn, Fe, Co, Ni, Cu, Nb, Rh, Pd, Ag, Ta, W, Re, and Au DACs. Then, $∆G_{H^{*}}$ corresponding to the most favorable adsorptions are shown in Figure 6 (details in Appendix A). A total of 13 TM elements such as Ti, Mn, Fe, Mo, Rh, Ag, Hf, W, Re, Os, Ir Pt, and Au, are predicted to have better HER activities in DACs than SACs. The results show that the DACs have better HER activities than SACs mainly in the cases where the bonding of the H atom to the TM atom is weakened in DACs. Re_2_−*SVGN*_3_ gives a $∆G_{H^{*}}$ value of −0.09 eV, which is the closest to zero among all the DACs and much closer to zero than Re_1_−*SVGN*_3_. On the contrary, for other elements, the catalytic activity of HER will be reduced if the TM atoms form DACs. The two elements of Co and Ni exhibit excellent HER catalytic activity in the SACs, which is consistent with previous study [22,38]. In addition, we found that Pd_1_−*SVGN*_3_ can be employed as a potential element for catalyzing HER, with the $∆G_{H^{*}}$ value of −0.09 eV. These results indicate the catalytic activity of these systems can be adjusted by delicately tuning the formation of SACs or DACs.

The intensity of H adsorption energy that is associated with charge transfer can directly affect the Gibbs free energy of H adsorption [20,52], and the Bader charge analysis can visualize charge transfer. To further investigate the HER activity of Co, Ni, Pd, and Re elements in SACs and DACs, the Bader charge analysis is performed with and without the H adsorption. As shown in Table 4, the total charge on the TMs increases upon the hydrogen adsorption. For Co, Ni, and Pd, the SACs lose fewer electrons than DACs upon H adsorption, resulting in the lower H adsorption capacities. On the contrary, the Re SAC loses more electrons (+0.35 e) than the DAC (+0.15 e), resulting in the weaker binding strength of the H atom on Re_2_. We also perform the charge density difference of Re_1_−*SVGN*_3_ and Re_2_−*SVGN*_3_ for before and after H adsorption (as presented in Figure 7), in order to further analyze the change of charge transfer during the catalytic process: (1) For Re SAC, before H adsorption, the Re single-atom is roughly positively charged due to the system’s electrical neutrality, electron cloud density focusing on the N atoms in the substrate as well as the Re single-atom, and the graphene substrate being negatively charged; after H adsorption, the Re single-atom is normally negatively charged due to the system’s electrical neutrality, electron cloud density focusing on the H atom and the Re single-atom, and the H atom being positively charged. (2) For Re DAC, before H adsorption, the Re dual-atom is normally negatively charged due to the system’s electrical neutrality, electron cloud density focusing on the N atoms in the substrate as well as the Re dual-atom, and the graphene substrate being positively charged; after H adsorption, the Re dual-atom is still negatively charged due to the system’s electrical neutrality, electron cloud density focusing on the H atom and the Re dual-atom, and the H atom being positively charged. In conclusion, we intuitively find more charge transfer for Re SAC, which is the same as our calculated results from Bader charge analysis. Thus, the smaller the electron loss upon H adsorption, the weaker the binding strength, which shifts $∆G_{H^{*}}$ towards the zero point in these cases. The different behavior of charge transfer in SACs and DACs determines the different activity.

## 4. Conclusions

In summary, by using DFT calculations, we systematically investigated the geometric structure, stability, and HER catalytic activity for the 3d, 4d, and 5d TM SACs and DACs on the *SVGN*_3_ substrate, respectively. We conclude the DACs of Mo, Tc, Re, Os, Ir, Pt, and Au are relatively more stable than the isolated SACs. The other DACs could also form when the TM atoms are slightly overloaded on the surface. Comparing the HER reaction overpotential, 13 DACs exhibit better catalytic performance than the SACs. After screening, Re_2_−*SVGN*_3_ is predicted to be good catalysts for HER. This study shows the importance of studying DACs for atomically-dispersed catalysis. It also provides useful information to design DACs on the *SVGN*_3_ surface for HER beyond SACs. Furthermore, we also found that both the electrochemical conditions and the lattice/matrix in which the metals are embedded affect the stability and HER activity of atomically-dispersed metal catalysts [51,53]. In future work, we will further consider the effect of the real environment on the stability and catalytic activity.

## Figures and Tables

**Figure 1 nanomaterials-12-02557-f001:**
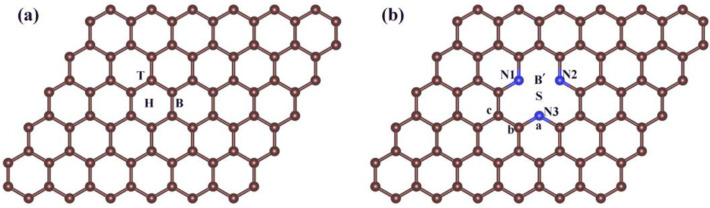
Color code: C, black; N, blue. (**a**) the three different adsorption sites (H (hollow), B (bridge), T (top)) in pure graphene; (**b**) in the SVGN3: a stands for the top position of nitrogen, b and c stand for the top positions of carbon at different positions; B’ stands for the bridge position between two nitrogen atoms; The symbol S represents the single vacancy in the substrate. The three nitrogen atoms are labeled as N1, N2, and N3, respectively.

**Figure 2 nanomaterials-12-02557-f002:**
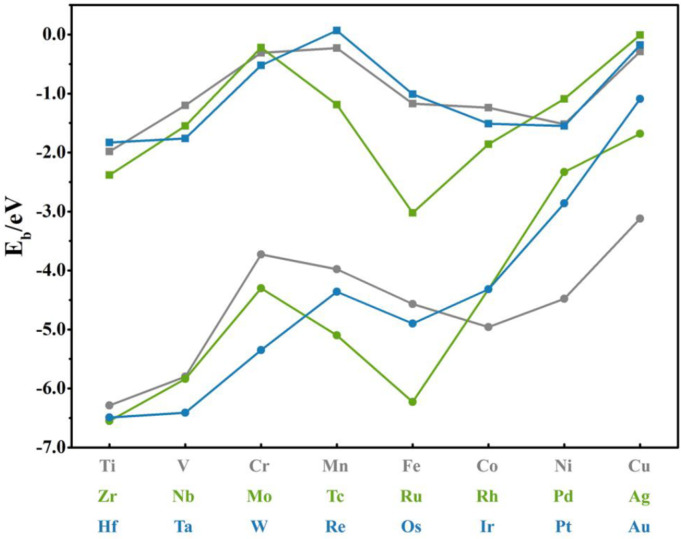
The binding energies of the *TM*_1_−*G* and *TM*_1_−*SVGN*_3_ are shown in square and circle, respectively.

**Figure 3 nanomaterials-12-02557-f003:**
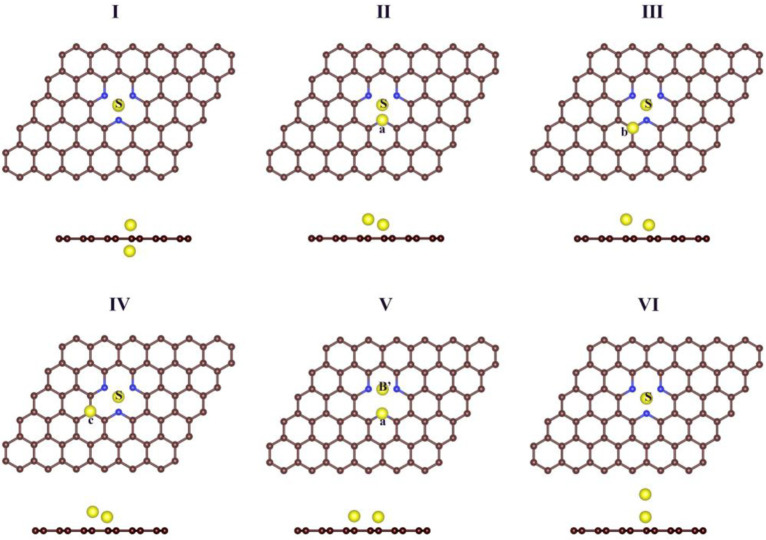
Color code: C, black; N, blue; TM, yellow. The six initial configurations are marked as (**Ⅰ**–**Ⅵ**).

**Figure 4 nanomaterials-12-02557-f004:**
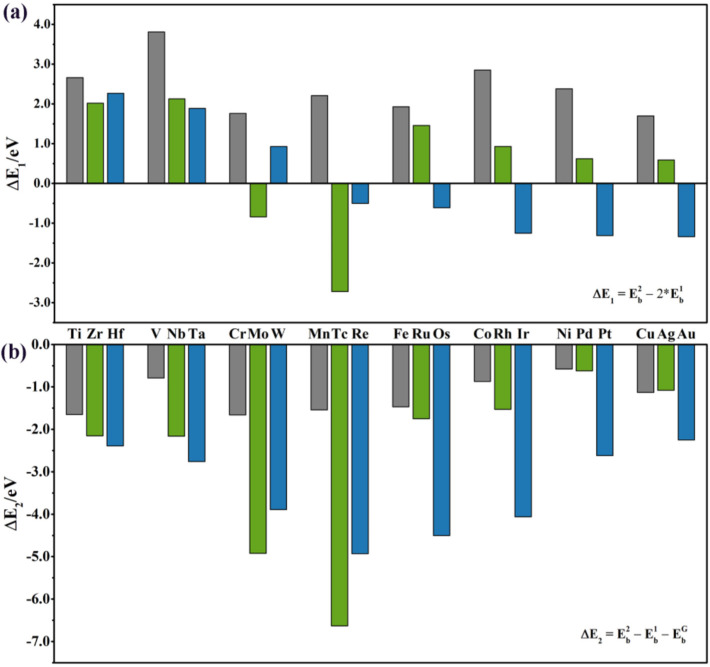
(**a**) $∆E_{1}$ is the difference between $E_{b}^{2}$ and $2*E_{b}^{1}$. (**b**) $∆E_{2}$ is the difference between $∆E_{2}$ and ($E_{b}^{1}+E_{b}^{G}$ ). (gray, green, and blue histograms represent for 3d, 4d, and 5d TM elements, respectively.)

**Figure 5 nanomaterials-12-02557-f005:**
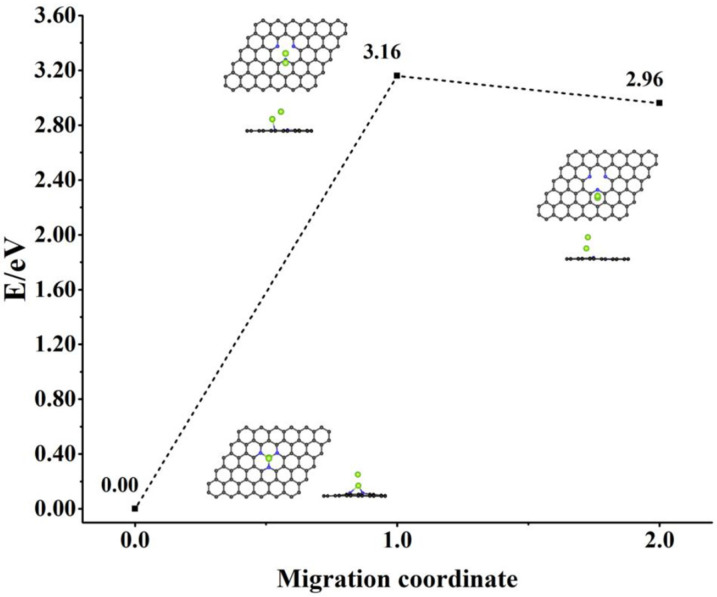
CI−NEB calculation for Re dual-atom migration in Re_2_−*SVGN*_3_. (Color code: C, black; N, blue; Re, green).

**Figure 6 nanomaterials-12-02557-f006:**
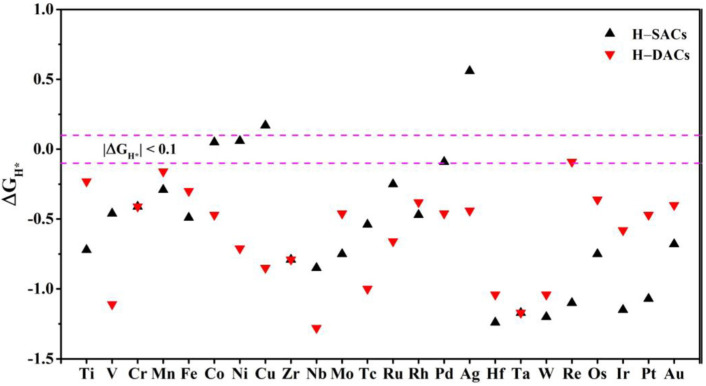
Gibbs free energy ($∆G_{H^{*}}$) for HER on the SACs is represented by the black triangle, corresponding $∆G_{H^{*}}$ on the DACs is represented by the red triangle.

**Figure 7 nanomaterials-12-02557-f007:**
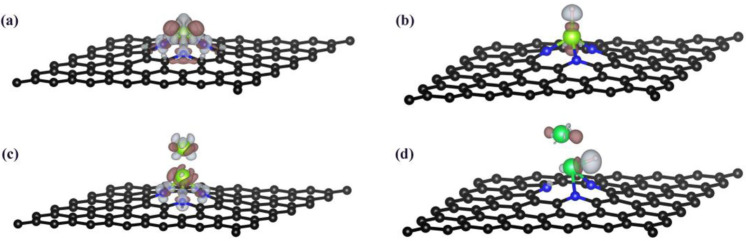
The charge density difference of (**a**,**c**) Re−*SVGN*_3_, Re_2_−*SVGN*_3_ without H adsorption and (**b**,**d**) Re−*SVGN*_3_, Re_2_−*SVGN*_3_ with H adsorption. (Color code: C, black; N, blue; Re, green; H, white; positive charge density, gray; negative charge density, dark red.).

**Table 1 nanomaterials-12-02557-t001:** The most stable adsorption site, binding energy ($E_{b}^{G}$ ), and M1 height from graphene (h) of *TM*_1_−*G*.

	Site	$E_{b}^{G}/\mathrm{eV}$	h/Å
Ti	H	−1.98	1.85
V	H	−1.20	1.85
Cr	B	−0.31	2.23
Mn	T	−0.23	2.24
Fe	H	−1.17	1.55
Co	H	−1.24	1.54
Ni	H	−1.52	1.56
Cu	T	−0.29	2.08
Zr	H	−2.38	1.98
Nb	H	−1.55	1.71
Mo	B	−0.22	2.32
Tc	H	−1.19	1.66
Ru	H	−3.02	1.73
Rh	H	−1.86	1.78
Pd	B	−1.09	2.06
Ag	B	−0.01	3.47
Hf	H	−1.83	1.93
Ta	H	−1.76	1.85
W	B	−0.52	2.22
Re	T	0.07	2.20
Os	H	−1.01	1.71
Ir	B	−1.51	1.94
Pt	B	−1.55	1.97
Au	T	−0.18	3.46

**Table 2 nanomaterials-12-02557-t002:** The most stable adsorption site, binding energy ($E_{b}^{1}$ ), the distance between the anchored metal atoms (M1) and N atom (d_M1−Nx_), and the M1 height from graphene (h) of *TM*_1_−*SVGN*_3_.

	Site	$E_{b}^{1}/\mathrm{eV}$	d_M1−N1_/Å	d_M1−N2_/Å	d_M1−N3_/Å	h/Å
Ti	S	−6.29	1.91	1.91	1.91	1.58
V	S	−5.80	1.90	1.90	1.90	1.52
Cr	S	−3.73	1.96	1.96	1.96	1.51
Mn	S	−3.98	2.00	2.00	2.00	1.65
Fe	S	−4.57	1.78	1.78	1.78	1.23
Co	S	−4.96	1.83	1.83	1.83	1.39
Ni	S	−4.48	1.84	1.85	1.85	1.37
Cu	S	−3.12	1.89	1.89	1.89	1.51
Zr	S	−6.55	2.07	2.06	2.06	1.84
Nb	S	−5.84	1.99	1.99	1.99	1.76
Mo	S	−4.30	1.94	1.94	1.94	1.68
Tc	S	−5.10	1.91	1.91	1.91	1.59
Ru	S	−6.23	1.93	1.93	1.93	1.55
Rh	S	−4.32	2.04	2.04	2.05	1.66
Pd	S	−2.33	2.27	2.11	2.11	1.78
Ag	S	−1.68	2.31	2.31	2.31	2.05
Hf	S	−6.49	2.00	2.00	1.99	1.75
Ta	S	−6.41	1.96	1.95	1.95	1.70
W	S	−5.35	1.91	1.91	1.91	1.65
Re	S	−4.36	1.91	1.91	1.91	1.60
Os	S	−4.90	1.91	1.91	1.91	1.55
Ir	S	−4.32	2.01	2.01	2.01	1.63
Pt	S	−2.86	2.09	2.24	2.09	1.73
Au	S	−1.09	2.34	2.34	2.35	2.03

**Table 3 nanomaterials-12-02557-t003:** The most stable configuration, binding energy ($E_{b}^{2}$ ), the distance between the anchored metal atoms (Mx) and N atom (d_Mx−Nx_) (M1 denotes the first metal atom, whereas M2 denotes a second attached atom), and the metal–metal distance of M1−M2 (d_M1−M2_).

	Configuration	$E_{b}^{2}/\mathrm{eV}$	d_M1−N1_/Å	d_M1−N2_/Å	d_M1−N3_/Å	d_M2−N1_/Å	d_M2−N2_/Å	d_M2−N3_/Å	d_M1−M2_/Å
Ti	Ⅰ	−9.92	2.08	2.10	2.10	2.15	2.06	2.06	2.82
V	Ⅱ	−7.79	1.95	1.95	2.04	3.48	3.48	1.98	2.03
Cr	Ⅵ	−5.70	2.07	2.07	2.07	3.26	3.27	3.28	1.51
Mn	Ⅵ	−5.75	1.96	1.96	1.96	3.82	3.78	3.81	2.28
Fe	Ⅵ	−7.21	1.85	1.85	1.85	3.43	3.44	3.43	2.06
Co	Ⅵ	−7.07	1.94	1.95	1.95	3.80	3.59	3.43	2.11
Ni	Ⅵ	−6.58	1.93	1.93	1.92	3.65	3.65	3.64	2.16
Cu	Ⅵ	−4.54	2.02	2.02	2.02	3.88	3.89	3.87	2.25
Zr	Ⅰ	−11.08	2.20	2.19	2.25	2.23	2.18	2.17	3.11
Nb	Ⅱ	−9.55	2.07	2.07	2.14	3.77	3.77	2.09	2.34
Mo	Ⅵ	−9.44	2.27	2.27	2.27	3.68	3.68	3.69	1.68
Tc	Ⅵ	−12.92	2.15	2.15	2.15	3.17	3.16	3.17	1.28
Ru	Ⅵ	−11.00	2.01	2.01	2.01	3.69	3.71	3.68	2.10
Rh	Ⅳ	−7.71	1.97	2.04	1.97	3.20	2.86	4.50	2.51
Pd	Ⅳ	−4.04	2.20	2.12	2.13	3.39	4.62	2.96	2.64
Ag	Ⅵ	−2.77	2.39	2.39	2.40	4.72	4.74	4.73	2.63
Hf	Ⅰ	−10.71	2.16	2.16	2.19	2.19	2.16	2.16	3.07
Ta	Ⅱ	−10.93	2.04	2.04	2.12	3.81	3.81	2.12	2.40
W	Ⅴ	−9.77	2.04	2.04	2.82	3.25	3.25	2.19	2.11
Re	Ⅵ	−9.22	2.08	2.08	2.09	3.79	3.79	3.74	2.07
Os	Ⅵ	−10.41	2.02	2.02	2.02	3.75	3.75	3.75	2.14
Ir	Ⅵ	−9.89	2.03	2.03	2.04	3.85	3.85	3.85	2.21
Pt	Ⅵ	−7.03	2.08	2.08	2.16	4.05	4.07	4.07	2.33
Au	Ⅵ	−3.52	2.38	2.23	2.52	4.52	4.60	4.60	2.52

**Table 4 nanomaterials-12-02557-t004:** Bader Charges, Q(e), of the typical TMs in SACs and DACs with non−H and H adsorption. (“+” means electron loss; “−” means electron gain).

**Non−H**	**System**	**Q_M1_/e**	**Q_M2_/e**	**Q_M1+M2_/e**	
	Co-*SVGN*_3_	+0.82	−	+0.82	
	Co_2_-*SVGN*_3_	+0.73	−0.13	+0.60	
	Ni-*SVGN*_3_	+0.73	−	+0.73	
	Ni_2_-*SVGN*_3_	+0.64	−0.18	+0.46	
	Pd-*SVGN*_3_	+0.52	−	+0.52	
	Pd_2_-*SVGN*_3_	+0.47	+0.11	+0.58	
	Re-*SVGN*_3_	+1.00	−	+1.00	
	Re_2_-*SVGN*_3_	+0.97	−0.17	+0.80	
**H adsorption**	**System**	**Q_M1_/e**	**Q_M2_/e**	**Q_M1+M2_/e**	** ∆Q/e **
	Co-*SVGN*_3_	+0.95	−	+0.95	+0.13
	Co_2_-*SVGN*_3_	+0.77	+0.08	+0.85	+0.25
	Ni-*SVGN*_3_	+0.74	−	+0.74	+0.01
	Ni_2_-*SVGN*_3_	+0.69	+0.08	+0.77	+0.31
	Pd-*SVGN*_3_	+0.54	−	+0.54	+0.02
	Pd_2_-*SVGN*_3_	+0.52	+0.10	+0.62	+0.04
	Re-*SVGN*_3_	+1.35	−	+1.35	+0.35
	Re_2_-*SVGN*_3_	+1.06	−0.11	+0.95	+0.15

## Data Availability

The data presented in this study are available on request from the corresponding author.

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
