# Peer review of "Investigation of the Stability and Hydrogen Evolution Activity of Dual-Atom Catalysts on Nitrogen-Doped Graphene"

_nanomaterials, 2022, doi:10.3390/nano12152557_

Round 1

Reviewer 1 Report

This paper performed DFT calculations to systematically study and compare the stability of TM atoms on two-dimensional substrates as well as to predict their HER activities based on Gibbs free energy (∆??). 3N-doped single vacancy graphene (SVGn3) seems to be a more suitable substrate for SACs and DACs due to its stronger binding energy with TM atoms compared with pure graphene (G). The authors explained the most stable configurations under each circumstance in details, however, more discussion and evidence are desired for better understanding of the unusual trends/data listed in the paper. All in all, I would recommend the publication of this paper on Nanomaterials after addressing the following comments.

1. In Figure 1a and line 137-140, the authors mentioned that TM1 tends to adsorb onto different locations on pure graphene. What’s the possible reasons/ explanation behind this?

2. In Table 2, despite all the elements embedded in the position of S point, Pd and Pt seem to be embedded asymmetrically with one dM1-Nx much larger than the other two.

3. Why does IVB elements preferably take the stable configuration as I (line 169-170)? Despite the similarity in VIII elements like Fe, Co, Ni, their DACs’ configurations seem difference. For Fe and Ni, dM-N are evenly distributed while for Co, dM2-N are different for 3 N atoms.

4. There seems to be no relationship between the stability in SACs/DACs configurations and corresponding HER activity. Will H-adsorption affect the stability of the SACs or DACs? If stability is affected, how will HER activity change?

5. In line 261-272, the discussion on Bader charge analysis and its relationship with H adsorption/∆??  lacks reference support. 

Reviewer 2 Report

Joural: Nanomaterials

Ms. ID. : nanomaterials-1828288

Title: Investigation of the stability and hydrogen evolution activity of dual-atom catalysts on nitrogen-doped graphene

Zhou et al. performed a computational study of a large number of transition metals adsorbed on pristine graphene and embedded in an N3 vacant site (SVGN3 site) in graphene lattice. The Authors also considered diatom (dimer formation) at the SVGN3 site at different configurations. Then, the studied systems were addressed in terms of stability and activity toward hydrogen evolution reaction. This is certainly an interesting work, and the methodology seems to fit the study goals. Also, the work fits well with the scope of the Journal. I consider that the manuscript is suitable for publication, but also believe that some important improvements are needed. The list of specific issues that should be addressed is listed below.

- To clearly show that the systems are set in the right way and that the data are reliable, a more detailed comparison of binding energies with literature data is needed. For example, the ref. 38 uses LDA approximation and it is difficult to compare the data in the present work with that study. More recent systematic works are available, like https://doi.org/10.1016/j.apsusc.2017.12.046.

- The SVGN3 site is rarely investigated to these days. Much more work is available for the so-called M@N4 catalysts. It would be interesting to compare the benefits of the SVGN3 site compared to the SV site in graphene lattice. The comparison with the data presented in https://doi.org/10.1039/C7CP07542A would be beneficial for understanding the role of N dopants in the stabilization of TM embedded in the N-doped graphene lattice.

- The Authors have used hydrogen adsorption energy as the descriptor of HER activity. This is a long-standing approach, but, strictly speaking, valid for extended surfaces. Moreover, taking only hydrogen binding energies as HER activity descriptor means that an acidic environment is considered, as in alkaline media water dissociation also impacts HER. Thus, stability under vacuum conditions can differ greatly from the stability under electrochemical conditions, and the Authors should also consider stability under electrochemical conditions, i.e., the dissolution of SACs and DACs. This can be done in different ways, and one recent work on the stability of metal SACs, https://doi.org/10.1016/j.electacta.2022.140155, could help Authors to address this issue.

- One more important point concerning SACs, DACs, etc. (highly dispersed metals) is that the lattice/matrix in which metals are embedded also contributes to the activity. The HER mechanism considered by the Authors is, as stated above, for extended surfaces. More detailed analysis/discussion is needed to address HER on SACs/DACs in more detail, see https://doi.org/10.1002/admi.202001814

Reviewer 3 Report

This is an interesting paper on single atom catalyst, conducted by Zhou and coworkers. Authors have calculated the adsorption energy and Bader charges of a ranges of systems of N-doped graphene, among a few other properties. However, I do not see any correlation between the between adsorption energy and vibration energy, which should be important in this kind of research. The effect of charge transfer on the catalytic process is not thoroughly discussed. Moreover, authors have mentioned that the Climbing image Nudged Elastic Band (CI-NEB) method was used to study the minimum energy path for dual-atom migration, but I cannot see any kind of PES in the paper. It is not clear what is a 6X6 supercell, and what is a 3X3X1 k-point? where X refers the alphabet “X”? The geometrical details of the systems examined are not discussed. Is there any relationship found between bond length and adsorption energy?  I ask the authors to fully address these issues in the revised version of the paper.  

Round 2

Reviewer 2 Report

The Authors have improved their work during the revision process and I believe that the manuscript can now be accepted without further modifications.

Reviewer 3 Report

Authors have revised their paper. However, I still have some concerns that the authors of the study should consider before I recommend the work for possible publication. 

1 - Fig. S2 is not clear. Please enlarge each figure in Fig. S2 and name each of them and then cite them in the main body of the paper. 

Details of computation of deltaG is not clarified in the computational section of the ms. Authors have just provided equations, but how they got the numerical values to calculate deltaG is not clarified. How was entropy deltaS calculated? Did they use DFPT? All technical details should be clarified. I ask the authors to provide this either in the main ms, or as supplementary info. Moreover, the authors should provide an example as supplementary info, showing how did they calculate deltaG. This will be helpful to anyone who wants to reproduce the data presented in this ms. Excel files that used for all calculations regardless of adsorption energy, Bader charges, etc., should be provided as supplementary.  
